# Healthcare Professionals’ Perceptions of and Attitudes towards a Standardized Content Description of Interdisciplinary Rehabilitation Programs for Patients with Chronic Pain—A Qualitative Study

**DOI:** 10.3390/ijerph20095661

**Published:** 2023-04-27

**Authors:** Douglas Anderson Åhlfeldt, Linda Vixner, Britt-Marie Stålnacke, Katja Boersma, Monika Löfgren, Marcelo Rivano Fischer, Paul Enthoven

**Affiliations:** 1Department of Health, Medicine and Caring Sciences (HMV), Linköping University, SE-581 83 Linköping, Sweden; 2School of Health and Welfare, Dalarna University, SE-791 88 Falun, Sweden; 3Department of Community Medicine and Rehabilitation, Umeå University, SE-901 87 Umeå, Sweden; 4The Center for Health and Medical Psychology, School of Law, Psychology and Social Work, Örebro University, SE-701 82 Örebro, Sweden; 5Division of Rehabilitation Medicine, Department of Clinical Sciences, Karolinska Institutet, SE-171 77 Stockholm, Sweden; 6Department of Rehabilitation Medicine, Danderyd University Hospital, SE-182 88 Stockholm, Sweden; 7Department of Health Sciences, Research Group Rehabilitation Medicine, Lund University, SE-221 00 Lund, Sweden; 8Department of Neurosurgery and Pain Rehabilitation, Skåne University Hospital, SE-221 85 Lund, Sweden

**Keywords:** interdisciplinary pain rehabilitation, content description, TIDieR checklist, healthcare professionals, complex interventions, individual interviews

## Abstract

Interdisciplinary pain rehabilitation (IPR) is a recommended treatment for people with chronic pain. An inadequate description of the content of IPR programs makes it difficult to draw conclusions regarding their effects. The purpose of this study was to describe the perceptions and attitudes of healthcare professionals toward a content description of IPR programs for patients with chronic pain. Individual interviews with healthcare professionals (*n* = 11) working in IPR teams in Sweden were conducted between February and May 2019. Analysis of the interviews resulted in a theme: interdisciplinary pain rehabilitation is a complex intervention, with three categories: limitations in the description of IPR programs; lack of knowledge about IPR and chronic pain; and facilitating and hindering factors for using the content description of IPR programs. Conclusion: Healthcare professionals perceived that IPR programs could be described through a general content description. A general content description could enhance the quality of IPR programs through a better understanding of their content and a comparison of different IPR programs. Healthcare professionals also expressed the importance of a content description being a guide rather than a steering document.

## 1. Introduction

Chronic pain, defined as pain that persists or recurs for more than three months [1], is a major cause of suffering and disability at the individual level. In Europe, the prevalence of chronic pain is about 19% [2]. Chronic pain is associated with poor health outcomes [3,4] and contributes to high socioeconomic costs as a result of reduced work-related productivity due to sick leave and healthcare costs [4,5,6]. The occurrence of pain has been reported in 30% of patients seeking primary care in Sweden [7], while even higher numbers (40%) have been reported in primary care in Finland. Half of the overall population seeking primary care due to pain in Finland suffered from chronic pain [8].

The etiology of chronic pain is thought to be multifactorial and has been associated with biomedical, psychological, and social factors [9]. The need for an interdisciplinary team and the use of a biopsychosocial approach [10] have been widely accepted as underlying principles for the treatment of patients with chronic pain [9]. Interdisciplinary pain rehabilitation (IPR) programs have proven to be more effective than usual care for patients with chronic pain for a variety of outcome measures such as pain, disability, quality of life, and sick leave [11,12,13,14,15]. According to the International Association for the Study of Pain (IASP), an IPR program is a multimodal treatment provided by a multidisciplinary team collaborating in assessment and treatment using a shared biopsychosocial model and goals [16]. The principal components of an IPR program are as follows: (1) a team assessment of the chronic pain problem and its consequences; (2) the establishment of a treatment plan, including interventions by different professions with goals to be achieved during the program; (3) communication among team members and between the team, the patient, and significant others; (4) deliveries of the different synchronized interventions of the IPR program; (5) evaluation of the interventions; (6) documentation; and (7) a discharge process, including interaction with other stakeholders [16,17]. IPR programs can include a variety of components (social, psychological, educational, and physical), healthcare professionals, and combinations of treatments with different intensities [18,19]. Another aspect is that IPR programs are patient-oriented, and individual rehabilitation plans are outlined according to the needs of each patient [16]. The evidence so far diverges regarding the importance of different individual characteristics for treatment outcomes. Results from one study of a large national registry favored women with university education [20], while another indicated that those in the worst situation regarding characteristics, such as psychological distress, low education level, and high pain severity, showed the largest improvements [21].

Several international studies have confirmed great variation in the content and design of treatment in IPR programs [18,19,22]. In a study by Rivano et al. [23], reports from different clinics (*n* = 31) providing IPR programs in Sweden indicate probable variation in clinical experience and ways of delivering care. The organization, content, and intensity of IPR programs revealed, for example, that patients treated per year ranged from three to three hundred and forty, program length ranged from 20 to 180 h, and only about 50% of reporting clinics had the same five healthcare professions in the IPR team (physiotherapist, occupational therapist, psychologist, physician, and social worker) [23]. These differences between the programs make it difficult to draw conclusions about the effect of IPR interventions. In addition, the fact that there is no detailed description of interventions within IPR programs makes it even more difficult [11,18,24].

The Swedish State and its municipalities and counties have agreements for funding treatment programs for chronic pain [25]. The purpose of these agreements is to stimulate access to IPR and other treatments for the population of working age who suffer from chronic pain. Furthermore, the aim is to have an effective process for sick leave and rehabilitation in order to stimulate a return to work [25]. Research has shown that IPR is an effective and evidence-based intervention for people with chronic pain to reduce sick leave and improve return to work [4,11,12,26,27,28]. The results are heterogeneous, with some patients showing improvement in a variety of areas while others do not improve at all [11,12,29,30]. In Sweden, both primary care and specialized care clinics provide interdisciplinary pain rehabilitation programs. Most specialized care clinics and a number of primary care clinics that provide IPR programs also report to the Swedish Quality Registry for Pain Rehabilitation (SQRP) [31]. The SQRP collects patient-reported outcome measures (PROM) through standardized questionnaires at baseline, after the intervention, and at 1-year follow-up [31]. Quality registries in the Swedish healthcare system serve the purpose of making it possible to evaluate the effects and potential risks of different treatments. Quality registries could also be used for comparing different healthcare providers/clinics in order to increase the quality of care [32].

To assess the quality of care given through different treatments, there are three critical components of information, according to Donabedian [33], namely, the structure of care, which includes organizational factors such as how healthcare personnel are organized and their level of expertise or education; the process of care, i.e., how care is provided, which components are included, and with what intensity the treatment is carried out; and finally, the outcome of care, in terms of pain intensity, health-related quality of life, or other measures that could be relevant to evaluate the treatment at hand. While the SQRP collects PROMS for patients receiving IPR, it does not collect information concerning the structure or the process within IPR programs, and thus, the quality of care of IPR programs in Sweden remains largely unknown.

There are no clear guidelines for how IPR programs should be designed, in terms of length and intensity of the rehabilitation program and combination of interventions, to achieve optimal effect [24]. For complex interventions such as IPR, a more structured description of components and content is necessary. Inadequate description of interventions limits the possibilities for healthcare personnel, patients, and other actors to draw conclusions about the results, and therefore their clinical use can be questioned [34]. In the process of creating a template for the content description of IPR programs, it would be beneficial to engage healthcare professionals who are actively engaged in IPR and therefore have specific knowledge of the design and content used. Little is known about healthcare professionals’ perceptions of and attitudes toward the use of a content description for IPR programs. The purpose of this study was therefore to describe the perceptions and attitudes of healthcare professionals toward a content description of interdisciplinary pain rehabilitation programs for patients with chronic pain.

## 2. Materials and Methods

This study is part of a research project with the aim of creating and piloting a national template for the content description of IPR programs in Sweden. The study was approved by the Regional Ethical Review Board in Linköping, Sweden (Dnr: 2019-01501) and conducted in accordance with the Declaration of Helsinki of 1964.

### 2.1. Study Design

Individual interviews were performed with healthcare professionals working with IPR for patients with chronic pain in different parts of Sweden to investigate their perceptions of and attitudes toward a content description for IPR programs. The interviews were analyzed by qualitative content analysis [35]. The study is reported in line with the Consolidated Criteria for Reporting Qualitative Research (COREQ) checklist for qualitative studies [36]; see Appendix A.

### 2.2. Subjects and Setting

Participants were recruited from specialist pain clinics or primary care in different counties in different parts of Sweden. Participants had to be actively working in an IPR program for the treatment of patients with chronic pain that consisted of at least two healthcare professionals from different professions. Physiotherapists, occupational therapists, psychologists, physicians, nurses, counsellors, and social workers are healthcare professionals commonly represented in IPR healthcare teams in Sweden. Purposive sampling was applied to provide variability in location, profession, extent of experience, age, and gender. All participants who were asked to participate in an interview accepted; there were no dropouts.

The IPR clinics included in the study had at least one member of an IPR team participate in an interview. The IPR teams varied in size, although all teams had an occupational therapist and a physical therapist. Two men and nine women from six different IPR teams in the counties of Dalarna, Uppland, Södermanland, Västerbotten, and Östergötland agreed to participate in the interview study. Most informants (this term is the equivalent term to interviewees and will be used exclusively in this article) had long experience working in IPR programs in either primary or specialized care. See Table 1 for background information about the informants.

### 2.3. Data Collection

The Template for Intervention Description and Replication (TIDieR) checklist and guide were developed by Hoffman et al. [34] and intended to facilitate the description of IPR programs in research studies [37], but the template can also serve as a tool for describing IPR programs at the national level in Sweden. The steps taken in creating a standardized content description followed a methodological framework as described by Moher et al. [38]. The development of the template was guided by current literature on the description of IPR programs [39,40], earlier attempts at using the TIDieR checklist as a descriptive tool [37,41], descriptions of IPR programs from IPR clinics in the counties of Västerbotten and Östergötland, Sweden, and unpublished material on checklists for the description of IPR programs provided by the co-authors LV and MRF. The structure of the template was inspired by the TIDieR checklist [34]. The first version of the content description included the following items: (1a–b) the name of the IPR and target population; (2) the program goal; (3a–3d) the material used; (4) clinical activities; (5a–j) background characteristics and clinical and academic experience of the provider of interventions; (6a–b) how the intervention is delivered; (7a–b) where the intervention is delivered; (8a–c) program length and intensity; (9) individual tailoring; and (11a–c) a description of compliance with the IPR program. A first draft of a template for the content description of IPR programs was created in January 2019 by DAÅ and PE. The draft was discussed by all the authors and four additional researchers in the field of chronic pain. The group included researchers and experts in chronic pain from different parts of Sweden and with different professional backgrounds, such as occupational therapists, physiotherapists, physicians, and psychologists, which resulted in the first version of a template for the content description of IPR programs in February 2019.

Information about the study and an invitation to participate were conveyed by email to healthcare professionals working in or managing IPR clinics. An interview guide inspired by Kallio et al. [42] was developed; see Appendix A. The interview guide contained questions on two areas: informants’ attitudes and perceptions regarding the first version of a standardized content description for IPR programs and informants’ attitudes and perceptions regarding the description of IPR programs. The interview guide was tested in a pilot interview. As a result of the pilot interview, the information on how to interpret the use of the content description was revised to emphasize more clearly that healthcare professionals were the primary users of the content description. This change did not lead to significant changes in the interview guide itself, so the pilot interview was included in the study. At the start of each interview, the interviewer asked questions about background data such as age, gender, profession, and work experience (see Table 1). One week before the interview was performed, the informants received the first version of a standardized content description for IPR programs via email with the instruction to read the content description in its entirety. In the email, the informants also received information about the study, including the fact that participation was voluntary and could be discontinued at any time. In addition, they were informed that the data would be handled in such a way as to be inaccessible to unauthorized personnel. All participants signed an informed consent form after receiving oral and written information.

The first author (DAÅ) conducted the interviews, which took place between February and May 2019. Interviews were conducted face-to-face at the interviewee´s workplace (*n* = 3), through an online video meeting program (*n* = 6), or via telephone (*n* = 2). The interviews lasted 40–68 min; they were audiotaped and transcribed verbatim. Field notes were taken during the interviews.

### 2.4. Data Management and Analyses

Qualitative content analysis [35,43] was applied to analyze the interview material. An inductive approach was applied, deriving a theme and categories from the data. First, the interviews were read in full to obtain an understanding of the text. Then the coding process started by identifying meaning units. Groups of meaning units were abstracted to a higher common meaning represented by a code [35]. Table 2 gives an example of the coding process. The first author (DAÅ) coded all interviews, and two other authors (LV and PE) each coded three interviews for triangulation. A discussion following the triangulation process between DAÅ, LW, and PE resulted in consensus. Subcategories and categories were created from groups of codes; all the authors were engaged in discussions on the sorting process into subcategories and categories. To assist the coding process, OpenCode 4.0 Umeå software was used (ICT Services and System Development and Division of Epidemiology and Global Health, Umea University, Sweden, 2015), which is available from https://www.umu.se/en/department-of-epidemiology-and-global-health/research/open-code2/ (accessed on 1 June 2019).

## 3. Results

The analysis of the interviews resulted in three categories, see Table 3. The first category, limitations of describing IPR programs, deals with difficulties in describing IPR given their complexity and individual tailoring. The second category, lack of knowledge about IPR and chronic pain, deals with the consequences of insufficient knowledge among colleagues, patients, and other stakeholders. The third category, facilitating and hindering factors for using the content description of IPR programs, deals with factors facilitated by the use of a content description, such as increased knowledge of the content and differences between IPR programs and guidance for continued development, and hindering factors, such as the risk of increased control and demands on the programs.

### 3.1. Limitations of Describing IPR Programs

According to informants, IPR programs contain several different treatments rooted in different theoretical perspectives. Informants described IPR as being complex and could see a number of difficulties that could hinder a reliable description of the IPR programs.

#### 3.1.1. Limitations of Describing the Complexity through a Content Description

IPR was described by informants as being complex with a rich content of different competencies, methods, and efforts. Informants also pointed out the difficulty of drawing clear boundaries between different treatments when, for example, the same treatment can be rooted in different theoretical perspectives.
*“Because certain things are done from a theoretical perspective by a therapist and then another therapist comes in and does the same things but has a different theoretical perspective and then suddenly calls it something else. Are you with me? So, one says this is ACT I am working with, and another says yes, it is ACT, but it is also operant therapy and exposure.” **[I:8]*

There was some skepticism among informants about the content description’s ability to describe the complexity of IPR. They expressed that the patients they meet have a complex of problems, rarely clear-cut diagnoses, and often co-morbidity, where one condition, for example, mental illness, can make another condition worse, for example, the pain problem. A complex of problems in patients is handled through a combination of interventions by different professions working together in teams. The result of the IPR in the form of positive effects cannot be solely derived from specific components; it rather reflects the coordination and combination of treatment efforts in the program.
*“… coordinated efforts of several professions that collaborate around and with a patient, where you work to achieve a defined goal formulated together with and by the patient.” “That we have a common foundation to stand on, that we all undergo common competence development, that we have a common point of view…” **[I:3]*

#### 3.1.2. Limitations of Describing Individual Tailoring

Informants expressed that it was important to adapt the treatment to the individual’s needs. Informants’ descriptions of IPR programs emphasized the fact that they were individually tailored within the framework of group-based programs. According to informants, the patient’s own experience of their needs was important to create active participation, ownership, motivation, and drive. Individual adaptation could be perceived by informants as a complicating factor in the description of IPR programs. Informants held the opinion that while it is possible to describe what an IPR program is supposed to contain at the group level, it would be more difficult to write a content description for an IPR program after individual adaptations.
*“That’s how you want to work at individual level, and I think we can’t describe the program (at individual level), but we should still be able to describe the whole, what a coordinated interdisciplinary program is, what we are aiming for, and which components it contains…” **[I:9]*

Informants expressed that it was important that the treatment be evidence-based and that the evidence describe treatment at the group or community level. The patients often have a complex problem, frequently with comorbidity, which is not always represented in research. The informants described the dilemma of balancing treatment plans that are supported by evidence but that, at the same time, should be adapted to the unique needs of the individual patient. Furthermore, informants expressed that the treatment goal could vary depending on whose perspective you take. As care providers, they were obliged to provide evidence-based care, and they received resources based on a partially politically determined objective to reduce sick leave and increase employment. The patient, on the other hand, may have a different goal that is more related to reducing or being able to manage their pain. According to the informants, the goal-setting within IPR should be based on the individual patient. However, the general goals of the IPR program should nevertheless be described.
*“I think that it (the IPR program) should be individually tailored and, at the same time, it should also follow the evidence and then it is more at group and community level so you end up with something in between. But it is clear that it must be individually adapted because it must motivate the individual, of course, it must be of help to the individual. (.) But it is hard. (.) It depends on which point of view you adopt. Is it society’s or the patient’s?” **[I:10]*

Informants expressed that it was important to nuance the description of goals with IPR; it is not just about reducing, for example, pain or fear and anxiety. It is also very much about increasing patients’ abilities and rediscovering a desire and motivation to move on. It is important that you, as a therapist, can provide the patient with tools and a buffer to cope with living with pain.
*“… what’s also important is when you have reduced the (patient’s) degree of depression and anxiety, I think one of my major tasks is to rediscover (the patient’s) desire and motivation to move on (.) you shouldn’t be back at zero when you leave here emotionally, so to speak, many enter (IPR) from the position “in the red”, so sometimes you stop when you are back, so to speak, at a neutral level, but if you are to endure a life with pain, then you need to have access to more strategies” **[I:5]*
*“Yes, and then it is pain rehabilitation to learn more about chronic pain and how you can manage the pain yourself, so more about the condition and self-coping strategies.” **[I:9]*

### 3.2. Lack of Knowledge about IPR and Chronic Pain

Informants felt that, generally speaking, there is insufficient knowledge about chronic pain in healthcare services. Informants described that patients they meet confirm the lack of knowledge about chronic pain in their previous contacts with other healthcare professionals, for example, in primary care. According to the informants, the patients themselves also need more information about how chronic pain can be managed and what interdisciplinary rehabilitation entails in more concrete terms. Even leading representatives for healthcare providers around the country have knowledge gaps about IPR, for example, regarding which patients may be suitable and the content of the treatment.

#### Lack of Knowledge Can Lead to Negative Consequences

Informants described how a lack of knowledge about chronic pain can lead to inadequate treatment for patients. Patients whom informants meet describe not being listened to or seen. According to the informants, there is a risk that healthcare professionals are thought to be able to solve everything, which in this case could be negative for patients who risk being passed around in the healthcare system. The fact that colleagues in other parts of the healthcare system do not have sufficient knowledge can lead to incorrect perceptions about which patients are suitable for treatment and also incorrect expectations about the outcome of the treatment.
*“…you think those who work in outpatient care (.) there, I feel that (.) I think I hear that patients feel they are not listened to nor seen (within outpatient care). And then especially our young adults “You who are so young cannot have so much pain” and that they (the patients) feel no affirmation and that they are not trusted.” **[I:4]*
*“Colleagues, healthcare managers who don’t understand what this implies. Who expect rehabilitation to work for the most difficult patients. What is often questioned is that we cannot help those extremely difficult patients who maybe need help with opioid tapering or psychiatric co-morbidity, where psychiatry is the main track really.” **[I:3]*

### 3.3. Facilitating and Hindering Factors for Using the Content Description of IPR Programs

Informants felt that a content description could be an adequate tool and support for describing the IPR program at their clinic on a general level. According to the informants, more knowledge about the content of different programs with opportunities to learn and compare clinics, together with an increased understanding of what one does, justified the creation of a content description. However, informants also thought that a content description could lead to increased demands from managers or other decision-makers in Swedish healthcare regarding evidence-based interventions, education of team members, and the number of different healthcare professionals, i.e., the quality of care. Informants expressed concern that higher demands and control of the quality of IPR could lead to smaller clinics shutting down and perhaps a greater exclusion of patients.

#### 3.3.1. A General Content Description Can Be Used

According to informants, a content description must be general in nature and set out the basic features of the program, i.e., its purpose, goals, and overall method. However, the balance between a general and specific description was difficult. With a general description, informants believed that the content could be described so that it applied to the majority of patients. The informants felt that the content description essentially captured the various parts of the IPR program.
*“…but then it becomes important, as I said, that the content description is so (-) general that you can promise that you (the patient) can actually get what is in the content description (.) but still so specific that it is comprehensible…” what exactly is included, and that is probably a difficult balance…” **[I:8]*

#### 3.3.2. Content Description as a Guiding Document for Increased Knowledge

The informants perceived that the content description could be a supporting or guiding document. The content description could contribute to increased clarity regarding the program’s content and structure, which in turn could increase knowledge and understanding of the differences between different IPR programs. The Swedish National Registry for Pain Rehabilitation was highlighted as a tool for measuring the effect of rehabilitation programs today, but, according to informants, a content description could provide an additional dimension as it would then be possible to see the content of the different programs.
*“… I think it looks good (.) it still leaves quite a lot open so we can describe the content ourselves…” So that (.) it’s a guiding document but it’s still not too controlled.” **[I:7]*
*“I think being able to compare is of great value (.) if you think that you are comparing different IPR treatments, for example, to understand (.) in any case get some picture of what the differences might have been between these different treatments.” **[I:8]*

According to informants, a clearer description of IPR programs could lead to increased incentives for the further development of the programs. They felt that patients could also benefit from a clearer description of the content of the IPR programs. A patient-adapted description could be developed based on a content description to increase patient knowledge.
*“… I think that if it exists (the content description) for those who work with it, then it will be. I think that you will be able to inform the patients more clearly (.) What is included.” **[I:9]*

The informants expressed that a content description could help to highlight patients with chronic pain and contribute to the dissemination of a more concrete description of what IPR programs contain and which patients are suitable.

Increased awareness that IPR programs exist could reduce the fear of outpatient care providers who do not know how to deal with certain symptoms, for example, in young adults with chronic pain. A content description could also contribute to highlighting skill gaps among practitioners within IPR. It would also be clearer for the clinic itself to know what it does and what resources are put into the IPR programs.
*“Yes, a benefit for me is of course that it clarifies the areas we work with, which means that I can detect if I have competence gaps.” “…so it can be easier to shape one’s role” **(I:5)*
*I think you can make a patient-friendly version then or what to say” **[I:10]*

#### 3.3.3. Content Description Leads to Increased Demands and Control

According to informants, a content description could lead to various consequences that entail demands for change, such as higher requirements to include different healthcare professionals or demands regarding the level of expertise/education, and informants had concerns about how it could affect the future of the service and the willingness to invest resources. According to informants, the content description should be used as a supporting or guiding document and not as a steering document for the purpose of regulating how care should be delivered. With excessive control, there were concerns that, for example, patients would be excluded from treatment opportunities or that it would become more difficult to adapt the treatment to an individual level. The requirement for detail in the description of the delivery of care was also highlighted as an important issue. Informants believed that an overly detailed description of the IPR program’s content was not meaningful.

According to informants, a national content description of the IPR program could mean increased requirements for competence and evidence-based treatments. One concern expressed by an informant working in a rural area was that services in some locations would shut down rather than try to meet the new requirements.
*“That somehow, you come to the conclusion that this is too expensive, we cannot invest in this.” **[I:3]*

Informants expressed that it was important that the content description’s requirement for the level of detail must be meaningful, both with regard to the ability of the clinic to deliver the program described and considering the fact that certain parts of the program, for example, teamwork or certain underlying treatment theories, become difficult to describe on the basis of a given content template.
*“To say exactly how many hours you use ACT, for example, it probably doesn’t really work (.) because it permeates the entire treatment.” “…but really, I would like to say this permeates the entire treatment, a large part of the treatment, a small part of the treatment. Something like that, you know what I mean?” **[I:8]*

The participants expressed that if the content description led to increased control, there was an increased risk that specific groups of patients regarding diagnosis or comorbidity who are currently being offered IPR would be excluded because there is insufficient evidence for the treatment or because they were not allowed to adapt to the extent that would be required.

## 4. Discussion

The analysis of the interviews resulted in three main areas: Firstly, the informants expressed concerns about how to describe IPR programs in such a way as to give a fair description of the content. Secondly, they expressed the view that healthcare professionals who do not work with IPR have insufficient knowledge about IPR programs and chronic pain. Thirdly, the informants experienced that general content descriptions of IPR programs could have several advantages, such as serving as a tool for increased understanding of specific treatment programs and the comparison of the content of different IPR programs. They also thought that a content description could lead to increased demands on and control over the programs. The results could contribute to the management of chronic pain since general content descriptions of IPR programs may serve as a tool for the development of specific treatment programs and the comparison of the content of different IPR programs.

Healthcare professionals saw limitations in describing IPR programs due to their complexity, i.e., consisting of multiple clinical activities, involving an interdisciplinary team, and individual tailoring [44,45]. Breugelmans et al. [46] described an IPR program using the TIDieR checklist and found similar limitations as described in the current study. According to Breugelmans et al. [46], it is not possible to provide an exact, identically replicable description of an IPR program. In line with Breugelmans et al. [46], Smith et al. [37] conclude that the level of detail in describing IRP programs has clear limits due to the complexity of the intervention and that it is sensitive to the context [37]. Although there is no consensus definition of what complexity is, there are several known characteristics inherent in a complex system [47]. Because of the nonlinear combinations of multiple inputs in a system, complex processes often show unpredictable behavior. In contrast to a single cause generating a single effect that you can expect from a simple reductionistic system, complex systems have multiple causalities [47]. In line with the experiences of informants in the current study, Brown [48] describes chronic pain as having key characteristics of a complex adaptive system, such as nonlinearity, being constantly susceptible to change (adaptation), and “hidden attractors”, meaning that the likely point of outcome is not evident. Although much is being invested in the treatment of chronic pain and positive outcomes do occur, the pattern of the system remains elusive [48]. The way to effectively influence outcomes in complex systems is by balancing flexibility and control, giving preference to simple rules that can easily be applied to a local context, accepting diversity, and seeking local-level initiatives rather than central control [48]. In the current study, informants expressed the importance of safeguarding the flexibility and adaptability of IPR programs, for example, through individual tailoring. Moreover, informants have emphasized the importance of a content description being a guiding instrument rather than a steering one. Informants’ experiences in the current study showed similarities with theories on complex systems, which further highlights their relevance and must be taken into consideration when designing a content description.

The experience of healthcare professionals that a lack of knowledge could lead to negative consequences has also been found in previous interviews with healthcare professionals in primary care conducted by Stenberg et al. [49]. Healthcare professionals experienced criticism from colleagues because the treatment of patients with chronic pain required a lot of healthcare resources. The opinion of colleagues was thought to be the result of a lack of knowledge about chronic pain as a condition and its treatment. Furthermore, informants felt that there was seldom any plan put forth by healthcare management on how to prioritize IPR [49]. The results of the current study show that healthcare professionals still experience a lack of knowledge among colleagues and managers.

Healthcare professionals in the current study believed that a general content description that also has scope for a certain level of detail could bring greater clarity to the content of IPR programs. Some attempts have been made to improve the description of IPRs using the TIDieR checklist [37,46]. According to the authors [37,46], the TIDieR checklist was useful in providing a structured framework for a systematic and detailed description of clinical activities and therapy elements that comprise the IPR program. Breugelmans et al. [46] departed from the standard format of the TIDieR checklist and adapted it to be able to adequately describe the eligibility criteria and evaluation procedures of the IPR program [46]. Smith et al. [37] used the Consensus on Exercise Reporting Template (CERT) for exercise interventions [50] as an adjunct tool to the TIDieR checklist to further specify information such as type, dosage, intensity, and frequency of the intervention, and whether it requires supervision or individualization. Using the CERT in future research could provide additional direction for reporting and increase the possibility of replication of exercise interventions [50].

Informants believed that a content description could help clarify the content of IPR programs and be a tool for dissemination to different stakeholders, such as patients, other healthcare professionals, and management. Registries for chronic pain, such as the SQRP, include many patient-reported outcome measures. However, to interpret these data, better knowledge of the content of the IPR programs is needed [23]. When comparing two different interdisciplinary pain rehabilitation programs, Spinord et al. [39] concluded that differences in the design and content of IPR programs did not predict different effects in outcome (pain intensity, emotional functioning, and activity of daily life) between groups, although it might have clinical relevance for different subgroups, in this particular study, different age groups, and gender [39]. Tseli et al. [51] evaluated IPR programs (*n* = 15) in Sweden and found no clinically significant differences regarding effectiveness in programs of different durations, indicating that an IPR program with a short duration could be as effective as a longer program. The results could contribute to the management of chronic pain since general content descriptions of IPR programs may serve as a tool for the development of specific treatment programs and to compare the content of different IPR programs.

Healthcare professionals in the current study expressed that a timely content description could lead to more control concerning the content of IPR programs. They also thought it could lead to higher demands on the IPR teams, which could have both positive and negative effects, for example, the number of different healthcare professionals included and/or more evidence-based treatments. If this were the case, healthcare professionals would raise some concerns about the risk of excluding patients due to geography, because smaller clinics in rural areas with fewer resources could be less inclined to continue providing IPR or because of a reduction in individual tailoring in order to adjust for differences in the patient population. Stenberg et al. [52] conducted interviews with healthcare professionals working in IPR teams in primary care, which showed a feeling of frustration about the overall focus of IPR on increasing return to work and decreasing sick leave because it may reduce the importance of increasing quality of life at the individual level. Moreover, there is a risk of excluding patients with chronic pain who are not expected to return to work. Healthcare professionals expressed opinions about the goals of IPR that in part conflicted with healthcare legislation regarding equal care for all patients [52]. This shows that external factors exercise some control over the course of IPR programs, which sometimes conflicts with the needs of the individual patient, and this is similar to the experiences expressed by healthcare professionals in this study.

Individual interviews were seen as an adequate research method to describe healthcare professionals’ perceptions of and attitudes toward a content description of IPR programs [53]. The interviewer, a practicing physiotherapist and, at the time of the study, a Master’s student in medical science, had relatively little experience in conducting research interviews, which might be a weakness of the study. However, he received continuous guidance from an experienced supervisor (PE). In our opinion, the participants’ responses gave detailed descriptions of their different perspectives. Our opinion is based on the views of the authors of the current study, who all have extensive experience in research within the area of pain rehabilitation, including qualitative research methods. The interviews with the informants were conducted during working hours via online meetings. A strength of this study is that the informants represented different professionals, was from different parts of Sweden, and was from both primary and specialized care clinics delivering IPR. Purposive sampling was applied, enhancing variability in age, gender, experience, and professional background. The variation in the informants’ backgrounds strengthens both the credibility and the transferability of the study results [35]. The informants were quite unanimous in their perceptions and attitudes toward the content description of IPR programs. However, informants from primary care clinics expressed a greater lack of knowledge about chronic pain and its treatment compared to informants from specialist pain clinics. Informants from primary care clinics also perceived more clearly that the introduction of a content description could be a greater threat to existing activities within primary care. Explanations could be differences in staff and financial resources and the fact that specialist pain clinics have existed for a longer time and are part of the established care for patients with chronic pain, while primary care clinics for chronic pain have existed for a relatively short period, and informants may feel uncertain about future developments. Although there was a large variation in the informants’ backgrounds, the number of included IPR clinics were limited, and it cannot be ruled out that the perceptions and attitudes of certain healthcare providers might be different across the clinics. Fewer men than women participated in the interview study, as was expected due to the general lack of male healthcare professionals working in IPR [52]. The authors included in the current study had different professional backgrounds, representing those healthcare professions that are commonly represented in IPR teams (physical therapists, occupational therapists, psychologists, and physicians), and also had long experience working with IPR clinically and/or in research.

## 5. Conclusions

Healthcare professionals perceived that a general content description could enhance the quality of interdisciplinary pain rehabilitation programs through a better understanding of their content and comparison of different IPR programs. However, they also expressed concerns regarding a general content description, as it could lead to increased demands and control. Therefore, they emphasized that a content description should be a guiding rather than a steering document.

## Figures and Tables

**Table 1 ijerph-20-05661-t001:** Background data of the informants (*n* = 11).

Informants	
Age, years, Md ^1^ (25–75%)	47 (35–59)
Sex	Women, *n* = 9
	Men, *n* = 2
Years working in healthcare, Md ^1^ (25–75%)	20 (8–30)
Years working in IPR ^2^, Md ^1^ (25–75%)	10 (5–25)
Healthcare profession	Occupational therapist, *n* = 4
	Physiotherapist, *n* = 3
	General practitioner, *n* = 1
	Psychologist, *n* = 3
Work setting	Specialist pain clinic, *n* = 7
	Primary care clinic, *n* = 4

^1^ Md, Median; ^2^ IPR, Interdisciplinary Pain Rehabilitation.

**Table 2 ijerph-20-05661-t002:** Examples of codes and subcategories in the category “Limitations in the description of IPR ^1^ programs”.

Meaning Unit	Code	Subcategory	Category
It looks similar to a bunch of interventions stacked on top of each other.	It is difficult to include all parts of the program.	Limitations of describing the complexity through a content description.	Limitations of describing IPR ^1^ programs.
People who are not involved cannot see the importance of the process and the holistic perspective.			
Individual problem areas guide the choice of treatment.	It is not possible to describe IPR programs that represent actual content at an individual level.	Limitations of describing individual tailoring.	
It will be difficult to differentiate between the content description and the actual treatment included at the individual level.			
Our programs do not look the same; they can differ because of patients’ wishes.			

^1^ IPR, Interdisciplinary Pain Rehabilitation.

**Table 3 ijerph-20-05661-t003:** Overall theme, categories, and subcategories of the qualitative content analysis.

Interdisciplinary Pain Rehabilitation Is a Complex Intervention		
Limitations of describing IPR ^1^ programs	Lack of knowledge about IPR ^1^ and chronic pain	Facilitating and hindering factors for using the content description of IPR ^1^ programs
Limitations of describing complexity through a content description	Lack of knowledge can lead to negative consequences	A general content description can be used
Limitations of describing individual tailoring		Content description as a guiding document for increased knowledge
		Content description leads to increased demands and control

^1^ IPR, Interdisciplinary Pain Rehabilitation.

## Data Availability

The data presented in this study are available upon reasonable request from the corresponding author.

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
