# Peer review of "Healthcare Professionals’ Perceptions of and Attitudes towards a Standardized Content Description of Interdisciplinary Rehabilitation Programs for Patients with Chronic Pain—A Qualitative Study"

_ijerph, 2023, doi:10.3390/ijerph20095661_

Round 1

Reviewer 1 Report

This study provides some useful information of, and insight in healthcare professionals´ perceptions of and attitudes towards content description of interdisciplinary rehabilitation programs for chronic pain patients. Overall, this study is well-writen. However, I have still some small remarks.

Title

Please add ":a qualitative study"

Introduction

My initial thought after reading the introduction was that there is something missing regarding the "patient-tailored" aspect of IPR. This is also mentioned by the interviewees but I think its something that already should be mentioned in the introduction as it can partly explain why there is no clear description and no standardization.

Line 84-85: I wonder whether there is some indication of why some people do not respond well while others do? Are there some clear differences (in patients, interventions or other aspects) which could explain this? 

line 105-106: Clear guidelines on detailed aspects of an IPR make no sense and I am glad that this is also something that is represented in the results/ discussion. However, mentioning that the lack of detailed guidelines as part of the rationale for the study seems not appropriate. Is it possible to nuance this a little? I also wonder whether additional information can be added to the introduction regarding the "general guidelines" that are available.

Methods:

11 participants is ok for a qualitative study. However, considering that there are different professions and 6 different centra, 11 seems to be rather limited (even with purposive sampling). Variations in used IPR may be present, and the contribution and expectations of certain healthcare providers might be different accross the centra's. This mihght result in different attitudes and beliefs regarding the topic. A bigger sample would have been more desirable. Please add info regarding sampling saturation. Add to limitation?

line 186-187: "A consensus discussion following the triangulation process resulted in consensus." => Who participated in the discussion?

"However, he received con- tinuous guidance from experienced supervisors and, in our opinion, the interviews were rich in content." Can you add/explain on what "your opinion" is based? What do the authors mean with “rich in content”. Can you give some more information about the "experienced supervisors" (Who? What is their experience?)

Discussion

I was wondering if there were any discrepancies in the responses of the interviewees? Maybe it would be interesting to add and discuss these? How can these differences be explained?

Author Response

The authors would like to thank the reviewers for their valuable comments on our manuscript, and the editor for providing the opportunity to submit an improved version.
The manuscript has been modified taking into account all comments. Answers are provided below the Reviewers' comments. The lines mentioned are those of the revised version of the manuscript. Changes in the revised manuscript have been highlighted in yellow.

Reviewer 2 Report

Dear authors,

Congratulations on your work.

I have some comments and suggestions to help you improve the manuscript.

INTRODUCTION

(1) An earlier and clear definition of IPR would be useful for the reader.

(2) The paragraph between lines 111 and 117, regarding the TIDieR looks fitter in the methodology section. Furthermore, it's not very clear if and how your participants had access to this template and how is it composed of. Could you give more details (in the methods) on how you developed it and how is it structured?

MATERIALS AND METHODS

(3) Regarding the subjects and settings, you mention that you recruited in specialist clinics or primary care, but then in Table 1, you mention university hospitals. Could you clarify this? Also, don't you think that having different contexts (acute and primary care) introduces an important bias regarding pain assessment, as its nature and management will surely differ? Would you consider adding this to your limitations?

(4) In Table 1, you should include the sample size (n=??) in the header.

(5) In subheading 2.3 - Data collection, more details should be given about the first version of the content description. How's it structured and how was it developed?

(6) Details regarding confidentiality and informed consent are missing.

RESULTS

(7) Avoid discussing the results in this section. Clearly present the findings. For example, the first paragraph looks more like a Discussion.

DISCUSSION

(8) Avoid repeating results. The first paragraph is redundant.

(9) Any insights regarding future research?

(10) It's not clear how your results specifically contribute to the management of chronic pain. Could you please give more details and correlate the opinions you've explored with the main topic under study?

CONCLUSION

(11) It's a copy-paste from the Abstract. Clearly answer the research question. Somehow, the words "perceptions" and "attitudes" disappeared throughout your paper.

REFERENCES

(12) A total of 19 references (38%) have 10 or more years. If possible, update some of them.

Author Response

(The authors gave the same response as above.)

Round 2

Reviewer 2 Report

Dear authors,

Thank you for your work and for considering the suggestions I've sent.

Best regards